# Private Gradient Estimation is Useful for Generative Modeling

Bochao Liu
Institute of Information Engineering,
Chinese Academy of Sciences
Sch. of Cyb. Sec, UCAS
Beijing, China
liubochao@iie.ac.cn

Pengju Wang
Institute of Information Engineering,
Chinese Academy of Sciences
Sch. of Cyb. Sec, UCAS
Beijing, China
wangpengju@iie.ac.cn

Weijia Guo
Institute of Information Engineering,
Chinese Academy of Sciences
Sch. of Cyb. Sec, UCAS
Beijing, China
guoweijia@iie.ac.cn

Yong Li
Institute of Information Engineering,
Chinese Academy of Sciences
Sch. of Cyb. Sec, UCAS
Beijing, China
liyong@iie.ac.cn

Liansheng Zhuang
Sch. of Cyb. Sec, USTC
Anhui, China
lszhuang@ustc.edu.cn

Weiping Wang
Institute of Information Engineering,
Chinese Academy of Sciences
Sch. of Cyb. Sec, UCAS
Beijing, China
wangweiping@iie.ac.cn

Shiming Ge[*]
Institute of Information Engineering,
Chinese Academy of Sciences
Sch. of Cyb. Sec, UCAS
Beijing, China
geshiming@iie.ac.cn

## Abstract

While generative models have proved successful in many domains, they may pose a privacy leakage risk in practical deployment. To address this issue, differentially private generative model learning has emerged as a solution to train private generative models for different downstream tasks. However, existing private generative modeling approaches face significant challenges in generating high-dimensional data due to the inherent complexity involved in modeling such data. In this work, we present a new private generative modeling approach where samples are generated via Hamiltonian dynamics with gradients of the private dataset estimated by a well-trained network. In the approach, we achieve differential privacy by perturbing the projection vectors in the estimation of gradients with sliced score matching. In addition, we enhance the reconstruction ability of the model by incorporating a residual enhancement module during the score matching. For sampling, we perform Hamiltonian dynamics with gradients estimated by the well-trained network, allowing the sampled data close to the private dataset's manifold step by step. In this way, our model is able to generate data with a resolution of 256×256. Extensive experiments and analysis clearly demonstrate the effectiveness and rationality of the proposed approach.

[*]Shiming Ge is the corresponding author(geshiming@iie.ac.cn)

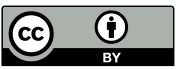

*MM '24, October 28-November 1, 2024, Melbourne, VIC, Australia*
© 2024 Copyright held by the owner/author(s).
ACM ISBN 979-8-4007-0686-8/24/10
https://doi.org/10.1145/3664647.3681713

## CCS Concepts

• **Security and privacy** → *Privacy protections*; • **Computing approachologies** → *Neural networks*.

## Keywords

Generative models, differential privacy, gradient estimation

**ACM Reference Format:**
Bochao Liu, Pengju Wang, Weijia Guo, Yong Li, Liansheng Zhuang, Weiping Wang, and Shiming Ge. 2024. Private Gradient Estimation is Useful for Generative Modeling. In *Proceedings of the 32nd ACM International Conference on Multimedia (MM '24), October 28-November 1, 2024, Melbourne, VIC, Australia.* ACM, New York, NY, USA, 9 pages. https://doi.org/10.1145/3664647.3681713

## 1 Introduction

Generative models have become indispensable tools across a broad spectrum of machine learning applications, such as image generation [4, 19, 21, 35, 36], text-to-image generator learning [2, 44] and imitation learning [27]. However, according to previous works [17, 58], synthetic data generated by these models could lead to data privacy leakage, as shown in Fig. 1. This issue has generated significant research interest in developing approaches to protect privacy without out reducing the usefulness of the generated data. The challenge is to find a careful balance between privacy and utility.

Differentially private (DP) [13, 14] generative modeling is an intuitive idea for addressing the challenge of privacy leakage, which trains DP generative models for privacy-preserving data generation. Many works [9, 20, 49, 56] adopt generative adversarial networks (GANs) [24] as the underlying generation backbone and incorporate the differential privacy into the training process, thereby bounding the privacy budget of the resulting generator. However,

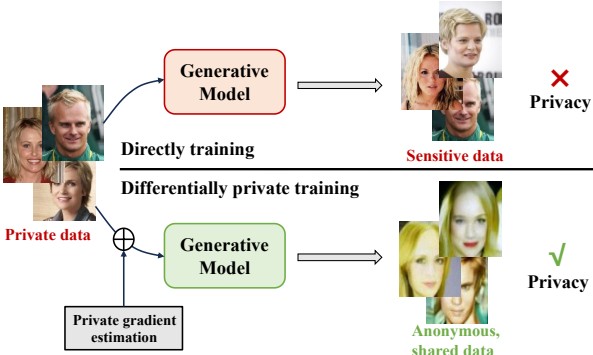

**Figure 1: Synthetic data generated by the generative model trained with private data directly may contain sensitive information. To address that, we achieve differentially private learning by private gradient estimation. Synthetic data generated by this generative model can be used for different downstream tasks with privacy protection.**

these GAN-based approaches rely on the assumption that the generator can generate the entire real records space to bootstrap the training process and are difficult to converge. Recently, with the potential of diffusion models [28] discovered, many works [10, 12, 39] have begun to explore how to train a privacy-preserving diffusion model. However, the extensive number of queries during training significantly compromises privacy. Consequently, these approachs necessitate thorough pre-training to minimize queries to private data. These challenges culminate in the inability to produce high-resolution images while preserving privacy.

Recent works in the field of generative modeling have underscored the promising capabilities of Energy-Based Models (EBMs) [33] for data generation. EBMs have been found to offer greater stability compared to GANs and require fewer queries to converge in comparison to diffusion models [46, 47]. This observation suggests that EBMs could serve as a solution to the challenges associated with generating high-dimensional data. Additionally, we have discovered that training EBMs with sliced score matching [45] effectively integrates with the randomized response (RR) [52] mechanism, which enables the achievement of differential privacy.

In this work, we propose the **P**rivate **G**radient **E**stimation (PGE) approach that learns to train a DP model to estimate the score of the private data, as shown in Fig. 1, and synthesize privacy-preserving images for downstream tasks. Instead of directly generating images, we train a network to estimate the gradient of logarithmic data density. Inspired by [26], we introduce a residual enhancement module that incorporates masked vectors, obtained by encoding $x$ with a pre-trained VQGAN [15], into the features extracted by the middle layer of the network $q_\theta$ to improve its reconstruction ability, as shown in Fig. 2. We adopt sliced score matching to train the network and achieve differential privacy by perturbing the projection vectors with RR. For sampling, as shown in Fig. 3, we design a Markov Chain Monte Carlo (MCMC) sampling approach based on Hamiltonian dynamics, which is more accurate than the commonly used sampling approach based on Langevin dynamics. In section 4

and supplementary material, we provide privacy and convergence analysis to support the effectiveness of our PGE further.

Our PGE approach effectively balances privacy and image quality through four key components. First, it replaces GANs with EBMs to better map sensitive data distributions, improving training stability. Second, we enhance differential privacy by using RR for perturbing projection vectors, which reduces randomness and boosts efficiency compared to traditional noisy addition approachs. Third, a residual enhancement module strengthens the network's ability to generate high-fidelity images. Finally, we use Hamiltonian dynamics-based MCMC sampling for more accurate image synthesis. Together, these elements create a solution that ensures both data privacy and high-quality image generation.

Our paper makes several key contributions as follows: (1) we propose the PGE approach, a differentially private generative modeling approach that effectively captures the distribution of private data while preserving valuable information. By leveraging Hamiltonian MCMC sampling, PGE can generate high-resolution images up to 256x256 with exceptional visual quality and data utility; (2) we introduce a residual enhancement module that can be flexibly applied to enhance the reconstruction capabilities of other generative models; (3) we conduct a comprehensive analysis of the privacy and convergence properties of PGE to validate its rationality and effectiveness; (4) experimental results demonstrate that, compared to other existing differentially private generative approaches, PGE significantly improves both the visual quality and data utility of the generated images.

## 2  Related Works

**Differentially private learning.** Differentially private learning aims to ensure the training model is differentially private regarding the private data. Existing approaches are typically based on differentially private stochastic gradient descent (DPSGD) [1, 6, 9, 56], which clipped and added noise to the gradients during the training process, and private aggregation of teacher ensembles (PATE) [38, 43, 49], which used semi-supervised learning to transfer the knowledge of the teacher ensemble to the student by a noisy aggregation. Recent works [10, 23, 40] applied randomized response (RR) [52] to the deep learning to achieve differentially private training. Despite significant progress in balancing data privacy and model performance, existing works are still far from optimal in the generative tasks. This is mainly because existing works apply training approachs for discriminative tasks directly to generative tasks. In contrast, we combine sliced score matching and randomized response well to realize differentially private generative modeling.

**Generative model learning.** With the development of generative techniques, recent works began to train generative models to generate data for downstream tasks. Recent works are typically based on GANs [24] and DDPM [28]. GAN-based approaches [11, 42, 57] are dedicated to improving training stability while improving the quality of the generated images. DDPM-based approaches [34, 41, 46] are committed to improving image generation quality while increasing generation speed. However, the instability of GANs and the high number of queries of DDPM make them difficult to generate high-resolution images under private training. Recently, some works [47, 55] have applied EBMs to generative tasks. It is more

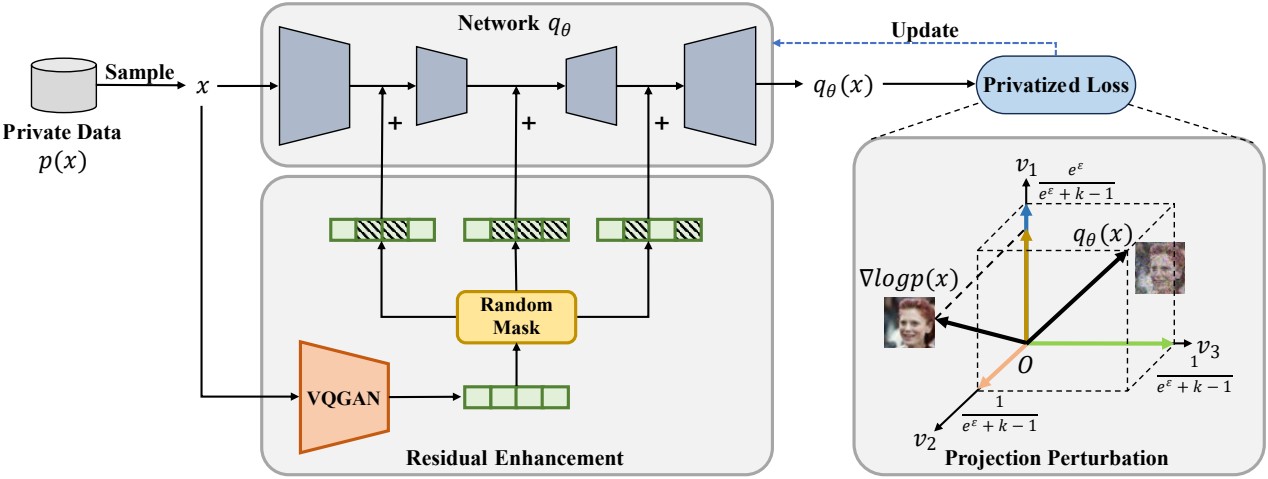

**Figure 2: Overview of our PGE. We first sample some images $x$ from the private data distribution $p(x)$. These images are then fed into the network $q_\theta$ for prediction. Concurrently, we encode these images using a pre-trained VQGAN and incorporate the masked version into the features extracted by the middle layer of $q_\theta$. This enhances the image reconstruction capability of $q_\theta$. Following the prediction by $q_\theta$, both $q_\theta(x)$ and $p(x)$ are projected for dimensionality reduction. During this process, we perturb their projection vectors by RR to achieve DP. Specifically, $\nabla \log p(x)$ is projected onto the $v_1$ direction, while RR projects $q_\theta(x)$ onto the $v_1$ direction with a probability of $e^\varepsilon/(e^\varepsilon + k - 1)$, and onto the other direction with a probability of $1/(e^\varepsilon + k - 1)$. Here, $k$ refers to the number of projection vectors. Finally, the network $q_\theta$ is updated by computing the loss between the predicted distribution $q_\theta(x)$ and the original distribution $p(x)$.**

stable than GANs and does not require as many queries as DDPM, which makes it more suitable for private generative modeling.

**Feature reconstruction.** Feature reconstruction is used in many domains and serves an important role. Masked autoencoder (MAE) [26] have emerged as a cornerstone technique that significantly enhances unsupervised feature learning by reconstructing images from masked inputs, showcasing improved efficiency in various image processing tasks. Following this, [16, 51] extended MAE to video processing, demonstrating reconstruction and feature extraction can be employed to enhance the temporal consistency of video frames. [5, 50] refined MAE applications in facial recognition, leading to advancements in both recognition accuracy and image quality. Face super-resolution [18, 29] can also be regarded as a form of feature reconstruction, where the model learns the mapping from low-dimensional data to high-dimensional data, reconstructing the missing information. Inspired by these works, we design a residual enhancement module to enhance both image quality and robustness of the model.

## 3 Preliminaries

**Energy-based models (EBMs).** EBMs capture dependencies by associating a probability density function to each configuration of the given variables. Given a known data distribution $T(x)$, we aim to fit it with a probabilistic model $E(\theta; x) = \exp(-H(\theta; x))/Z_\theta$, where $H(\theta; x)$ is an energy function with parameter $\theta$. As $E(\theta; \cdot)$ represents a probability distribution, it needs to be divided by a normalizing constant $Z_\theta = \int \exp(-H(\theta; x)) dx$. $Z_\theta$ is difficult to calculate explicitly, but as it happens, the image generation process in our paper only requires the gradient of logarithmic data density

$\nabla \log E(\theta; \cdot)$, which eliminates the need to compute it in our training. Notably, it is easy to extend to the multi-dimensional case as long as multiple variables are distributed independently of each other.

**Differential privacy (DP).** DP bounds the change in output distribution caused by a small input difference for a randomized mechanism. It can be described as follows: A randomized mechanism $\mathcal{R}$ with domain $\mathbb{N}^{|x|}$ and range $\mathcal{A}$ is $(\varepsilon, \delta)$-differential privacy, if for any subset of outputs $O \subseteq \mathcal{A}$ and any adjacent datasets $D$ and $D'$:

$$Pr[\mathcal{R}(D) \in O] \leq e^\varepsilon \cdot Pr[\mathcal{R}(D') \in O] + \delta, \tag{1}$$

where adjacent datasets $D$ and $D'$ differ from each other with only one training example. $\varepsilon$ is the privacy budget, where the smaller is better, and $\delta$ is the failure probability of the mechanism $\mathcal{R}$. In our case, the randomized response mechanism enables the EBMs to satisfy $(\varepsilon, 0)$-DP (or $\varepsilon$-DP). Notably, DP is featured by post-processing theorem and parallel composition theorem. The former could be described as: If $\mathcal{R}$ satisfies $(\varepsilon, \delta)$-DP, $\mathcal{R} \circ \mathcal{H}$ will satisfy $(\varepsilon, \delta)$-DP for any function $\mathcal{H}$ with $\circ$ denoting the composition operator. And the latter could be described as: If each randomized mechanism $\mathcal{R}_i$ in $\{\mathcal{R}_i\}_{i=1}^n$ satisfies $(\varepsilon, \delta)$-DP, then for any division of a dataset $D = \{D_i\}_{i=1}^n$, the sequence of outputs $\{\mathcal{R}_i(D_i)\}_{i=1}^n$ satisfies $(\varepsilon, \delta)$-DP regarding the dataset $D$.

## 4 Approach

This section outlines our PGE approach, discussing three key aspects. Firstly, we introduce how to fit a data distribution with an EBM privately and how to train a releasable network in practice. Secondly, we describe how to sample to obtain privacy-preserving

images. Lastly, we theoretically prove that our PGE can guarantee privacy and convergence.

## 4.1 Private Gradient Estimation

Physically speaking, we assume that the energy of the private data system is $T(x, c) = U(x) + K(c)$, where $x$ represents the position and $c$ represents the velocity. So there is:

$$
\begin{aligned}
p(x, c) &\propto \exp\left(-T(x, c)\right) \\
&= \exp(-U(x))\exp(-K(c)) \\
&\propto p(x)p(c),
\end{aligned} \tag{2}
$$

where $p(x)$ and $p(c)$ are canonical distributions of position $x$ and velocity $c$, and both are independently distributed. We find that the joint distribution can be sampled using the distributions of the random variables $x$ and $c$. To simplify the calculation, we assume that the distribution of the velocity $c$ is known and that the kinetic energy has:

$$
K(c) = -\log p(c) \propto \frac{c^T c}{2}. \tag{3}
$$

Similarly, the potential energy function can be expressed as $U(x) = -\log p(x)$.

**Manifold estimation with an EBM.** We use an EBM $E(\theta; x) = \exp(H(\theta; x))/Z_\theta$ to estimate the canonical distribution of energy function $p(x)$. As mentioned in the preliminaries section, we only need to estimate the gradient of logarithmic data density $\nabla \log p(x)$, so we build the loss function with Fisher divergence as follows:

$$
\begin{aligned}
&\mathcal{D}_F\left(p(x)\|E(\theta; x)\right) \\
&= \mathbb{E}_{x \sim p(x)}\left[\frac{1}{2}\left\|\nabla \log p(x) - \nabla \log E(\theta; x)\right\|^2\right] \\
&= \mathbb{E}_{x \sim p(x)}\left[\frac{1}{2}\left\|\nabla U(x) - \nabla H(\theta; x)\right\|^2\right].
\end{aligned} \tag{4}
$$

During the backpropagation process, it is necessary to compute the Hessian matrices of both $p(x)$ and $E(\theta; x)$. However, due to the high dimensionality of these matrices, the computational requirements are significant. To address this issue, we adopt the random projection approach proposed in [45] to reduce the dimensionality. First, we sample a projection vector $v$ from a standard Gaussian distribution. Then, we project the gradients of $U(x)$ and $H(\theta; x)$ onto the direction of the projection vector $v$. Finally, compute the loss function as follows:

$$
\mathcal{L}(\theta; x, v) = \mathbb{E}_{x \sim p(x)}\left[\frac{1}{2}\left\|v^\top \nabla U(x) - v^\top \nabla H(\theta; x)\right\|^2\right]. \tag{5}
$$

**Residual enhancement.** In practice, we train a network $q_\theta$ instead of $\nabla H(\theta; x)$ in Eq. (5) to fit $\nabla U(x)$ directly. Our approach can be understood as a process of reconstructing the image information, and inspired by masked autoencoder [26], we add a residual enhancement module, as shown in Fig. 2, to improve the reconstruction ability of the model. Specifically, for a batch of data $\{x_i\}_{i=1}^b$, we enter them into the model $q_\theta$ to predict the data manifold. In addition, the data is encoded by a pre-trained VQGAN, and the masked version is incorporated into the features extracted by the middle layer of the model $q_\theta$. This residual enhancement module improves the reconstruction ability of the model [26]. In the sampling process, as shown in Fig. 3, we add noise vectors sampled from a Gaussian

distribution to the features instead of masked features encoded by the VQGAN. Adding noise vectors improves the robustness of the model compared to adding nothing, especially when the data are located in low-density regions [47].

**Projection perturbation.** Training the network $q_\theta$ with the gradient of logarithmic data density $\nabla \log p(x)$ directly may introduce privacy risks. To address this concern, we perform RR to prevent leakage of private information. It perturbs the training data so that the trained network does not reveal the true distribution of private data during the sampling process. Therefore, we can release the trained network without concern about compromising privacy, as it is difficult for adversaries to tell whether an image is in the training data. Specifically, we aim to privatize the gradient of potential energy function $\nabla \log p(x)$ in Eq. (4), which is also equivalent to privatizing $v^\top \nabla U(x)$ in Eq. (5). For a batch of data $x = \{x_i\}_{i=1}^b$ and its projection vector $v = \{v_i\}_{i=1}^b$, there is an inherent correspondence between them, so the projection vectors $v_i$ of $\nabla H(\theta; x_i)$ and $\nabla P(x_i)$ are the same without any protection. In our case, we apply a RR mechanism $\mathcal{R}(\cdot)$ to perturb the projection vector of $\nabla U(x_i)$, which protects the private information by making the projection vectors of $\nabla U(x)$ and $\nabla H(\theta; x)$ not strictly aligned. This perturbation privatizes $v^\top \nabla p(x)$, which indirectly randomizes true directions toward perceptually realistic images. The RR mechanism $\mathcal{R}(\cdot)$ can be formulated as follows:

$$
\Pr\left[\mathcal{R}(v_i) = v_o\right] = \begin{cases} \dfrac{e^\varepsilon}{e^\varepsilon + k - 1}, v_o = v_i \\ \dfrac{1}{e^\varepsilon + k - 1}, v_o = v_i' \in v^- \setminus \{v_i\} \end{cases}, \tag{6}
$$

where $v^-$ is a subset of $v$, $k = |v^-|$ and $k \geq 2$. We choose the first $k$ projection vectors with the smallest cosine distance from $v_i$ to $v$ to reduce the impact of RR without compromising privacy. Compared to most other approaches with a small failure probability $\delta$, $\mathcal{R}(\cdot)$ achieves pure DP with a failure probability of 0. By this point, the loss function can be expressed as follows:

$$
\begin{aligned}
&\mathcal{L}(\theta; x, v, \mathcal{R}(\cdot)) \\
&= \mathbb{E}_{x \sim p(x)}\left[\frac{1}{2}\left\|\mathcal{R}(v^\top)\nabla U(x) - v^\top \nabla H(\theta; x)\right\|^2\right].
\end{aligned} \tag{7}
$$

Although we privatize $\nabla \log p(x)$, but we cannot obtain $\nabla U(x)$ in Eq. (7) directly. Usually, we do not know the exact form of the given data distribution. To address that, we assume that as training proceeds, $q_\theta(x)$ converges to $\nabla U(x)$ and both satisfy some weak regularization conditions as mentioned in [30]. So combined with [45], our loss function can be formulated as follows:

$$
\begin{aligned}
&\mathcal{L}(\theta; x, v, \mathcal{R}(\cdot)) \\
&= \mathbb{E}_{x \sim p(x)}\left[\mathcal{R}(v^\top)\nabla q_\theta(x)\mathcal{R}(v) + \frac{1}{2}\left(v^\top q_\theta(x)\right)^2\right].
\end{aligned} \tag{8}
$$

In this way, we can estimate the manifold of private data $p(x)$ with an EBM $E(\theta; x)$. In addition to this, we train the model with the perturbed data manifold estimation to ensure that the model conforms to DP. Overall, PGE can be formally proved $\varepsilon$-DP in Theorem 1. Detailed analysis can be found in supplementary material.

THEOREM 1. *Our PGE satisfies $\varepsilon$-DP.*

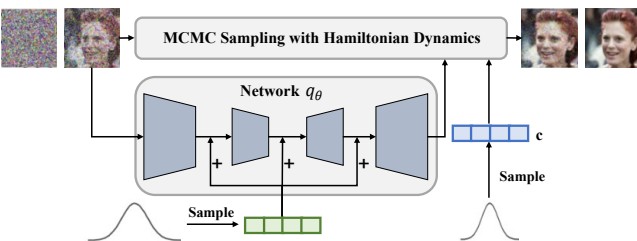

**Figure 3: Overview of the sampling process. Given a well-trained network, it can predict the gradients required by Hamiltonian dynamics. After several rounds of sampling, the samples gradually converge from a noisy distribution to the distribution of private data.**

## 4.2 Sampling with Hamiltonian Dynamics

After the training process is completed, we perform MCMC sampling with the leapfrog approach of Hamiltonian dynamics to generate images, as shown in Fig. 3. Following the previous section, for the distribution of private data $p(\boldsymbol{x})$, we have,

$$p(\boldsymbol{x}, \boldsymbol{c}) = p(\boldsymbol{x}) \cdot p(\boldsymbol{c}) \propto \exp(-H(\theta; \boldsymbol{x})) \cdot \exp(-K(\boldsymbol{c})). \quad (9)$$

And the sampling process could be described as follows:

$$
\begin{aligned}
\boldsymbol{c}(t + \frac{\lambda}{2}) &= \boldsymbol{c}(t) - \frac{\lambda}{2}\nabla_{\boldsymbol{x}} \log p(\boldsymbol{x}(t), \boldsymbol{c}(t)); \\
\boldsymbol{x}(t + \lambda) &= \boldsymbol{x}(t) + \lambda\nabla_{\boldsymbol{c}} \log p(\boldsymbol{x}(t), \boldsymbol{c}(t + \frac{\lambda}{2})); \quad (10) \\
\boldsymbol{c}(t + \lambda) &= \boldsymbol{c}(t + \frac{\lambda}{2}) - \frac{\lambda}{2}\nabla_{\boldsymbol{x}} \log p(\boldsymbol{x}(t + \lambda), \boldsymbol{c}(t + \frac{\lambda}{2})),
\end{aligned}
$$

where $\lambda$ is the step size. According to Eq. (9), we know $-\nabla_{\boldsymbol{x}} \log p(\boldsymbol{x}, \boldsymbol{c}) = \nabla_{\boldsymbol{x}}H(\theta; \boldsymbol{x}) = q_\theta(\boldsymbol{x})$ which is the output of the trained network and $-\nabla_{\boldsymbol{c}} \log p(\boldsymbol{x}, \boldsymbol{c}) = \nabla_{\boldsymbol{c}}K(\boldsymbol{c})$. According to Eq. (3), we know $\nabla_{\boldsymbol{c}}K(\boldsymbol{c}) = \alpha \cdot \boldsymbol{c}$. $\alpha$ is the scale factor, which could be combined into the step size $\lambda$, so we consider $\nabla_{\boldsymbol{c}}K(\boldsymbol{c}) = \boldsymbol{c}$. After simplification the sampling process is as follows:

$$
\begin{aligned}
\boldsymbol{c}(t + \frac{\lambda}{2}) &= \boldsymbol{c}(t) + \frac{\lambda}{2}q_\theta(\boldsymbol{x}(t)); \\
\boldsymbol{x}(t + \lambda) &= \boldsymbol{x}(t) - \lambda\boldsymbol{c}(t + \frac{\lambda}{2}); \quad (11) \\
\boldsymbol{c}(t + \lambda) &= \boldsymbol{c}(t + \frac{\lambda}{2}) + \frac{\lambda}{2}q_\theta(\boldsymbol{x}(t + \lambda)),
\end{aligned}
$$

In this way, the distribution of $\boldsymbol{x}(T)$ will converge infinitely to $p(\boldsymbol{x})$ when $T \rightarrow \infty$, in which case $\boldsymbol{x}(T)$ could be considered an exact sample from $p(\boldsymbol{x})$ under some regularity conditions [53]. Moreover, subsequent to executing the described sampling procedure for a predefined number of iterations ($N$), the Metropolis Criteria are employed to adjudicate the acceptance of the image's most recent state. This decision is governed by the acceptance probability, expressed as $\min(1, q_\theta(\boldsymbol{x}(t + N\lambda))/q_\theta(\boldsymbol{x}(t)))$. Such a mechanism ensures a thorough exploration of the entire distribution of private data, denoted as $p(\boldsymbol{x})$, thereby facilitating the generation of images with enhanced realism. Inspired by the learning rate adjustment technique in machine learning, we adopt the strategy of step size decay to speed up the sampling process. Initially, a larger step size is utilized to expedite the movement towards realistic

images. Subsequently, a smaller step size is employed to refine the image details. As an added benefit, because the starting step size is larger, it somewhat also prevents the model from collapsing into certain data-dense regions. Notably, during the sampling process, we no longer use VQGAN to encode the model's input but instead sample directly from a Gaussian distribution to add to the features extracted by the model. On one hand, it speeds up the sampling process compared to using VQGAN encoding, and on the other, it improves the robustness of the model, especially in increasing the diversity of generated data.

## 4.3 Convergence Analysis

In our convergence analysis, we primarily focus on the model parameters. The foundation of our analysis largely draws upon the methodologies outlined in [3]. We consider a worst-case scenario where the output and input of the RR mechanism differ, resulting in opposite directions of the gradients. Under this condition, applying algorithm $\mathcal{R}(\cdot)$ to the projection vectors effectively becomes equivalent to its application on the gradient. To better express its properties, we rewrite $q_\theta(\cdot)$ as $q(\theta; \cdot)$. Adhering to the same five assumptions in [3], we posit (1) $||\nabla q(\theta; \cdot) - \nabla q(\theta'; \cdot)||_2 \leq \kappa||\theta - \theta'||_2$; (2) $q(\theta; \cdot) \geq q(\theta'; \cdot) + \nabla q(\theta'; \cdot)^T(\theta - \theta') + \frac{1}{2}c||\theta - \theta'||_2^2$; (3) $\nabla q(\theta; \cdot)^T \mathbb{E}_{\boldsymbol{x}}[g(\theta; \boldsymbol{x})] \geq \mu||\nabla q(\theta; \cdot)||_2^2$; (4) $||\mathbb{E}_{\boldsymbol{x}}[g(\theta; \boldsymbol{x})]||_2 \leq \mu_G||\nabla q(\theta; \cdot)||_2$; (5) $\mathbb{V}_{\boldsymbol{x}}[g(\theta; \boldsymbol{x})] \leq C + \mu_V||\nabla q(\theta; \cdot)||_2^2$, where $\theta$ and $\theta'$ are the weights of model $q$, $\nabla q(\theta; \cdot)$ is the optimal gradient theoretically, $g(\theta, \boldsymbol{x})$ is the gradient we computed, $\mathbb{E}[\cdot]$ is the symbol for mean calculation, $\mathbb{V}[\cdot]$ is the symbol for variance calculation and $\kappa, c, \mu, \mu_G, \mu_V, C$ are non-negative constants. Based on these assumptions, we arrive at the following conclusions,

$$
\begin{aligned}
\mathbb{E}[q(\theta_{k+1}; \cdot) &- q(\theta^*; \cdot)] + \frac{\gamma^2 \kappa C}{4\tau c} \\
&\leq (2\tau c + 1)(\mathbb{E}[q(\theta_k; \cdot) - q(\theta^*; \cdot)] + \frac{\gamma^2 \kappa C}{4\tau c}),
\end{aligned}
\quad (12)
$$

where $\tau = -\gamma(\frac{2e^\varepsilon}{e^\varepsilon + k - 1} - 1)\mu + \frac{1}{2}\gamma^2\kappa(\mu_G^2 + \mu_V)$. When we guarantee that $\tau < 0$, it will converge and the error from the minimum $q(\theta^*; \cdot)$ is $-\frac{\gamma^2 \kappa C}{4\tau c}$. More details can be found in supplementary material.

## 5 Experiments

To verify the effectiveness of our proposed PGE, we compare it with 11 state-of-the-art approaches and evaluate the data utility and visual quality on four image datasets. To ensure fair comparisons, our experiments adopt the same settings as these baselines and cite results from their original papers.

## 5.1 Experimental Setup

**Datasets.** We conduct experiments on four image datasets, including MNIST [32], FashionMNIST (FMNIST) [54], CelebA [37] and LSUN [59]. We use the official preprocessed version with the face alignment and resize the images in CelebA to 64×64 and 256×256. CelebA-H and CelebA-G are created based on CelebA with hair color (black/blonde/brown) and gender as the label. For LSUN, we choose the bedroom category and resize the images to 256×256 to evaluate the perceptual scores.

**Table 1: Classification accuracy comparisons with 14 state-of-the-art baselines under different privacy budget $\varepsilon$.**

| | MNIST | | FMNIST | | CelebA-H | | CelebA-G | |
|---|---|---|---|---|---|---|---|---|
| | $\varepsilon=1$ | $\varepsilon=10$ | $\varepsilon=1$ | $\varepsilon=10$ | $\varepsilon=1$ | $\varepsilon=10$ | $\varepsilon=1$ | $\varepsilon=10$ |
| *Without pre-training* | | | | | | | | |
| **DP-GAN** (arXiv'18) | 0.4036 | 0.8011 | 0.1053 | 0.6098 | 0.5330 | 0.5211 | 0.3447 | 0.3920 |
| **PATE-GAN** (ICLR'19) | 0.4168 | 0.6667 | 0.4222 | 0.6218 | 0.6068 | 0.6535 | 0.3789 | 0.3900 |
| **GS-WGAN** (NeurIPS'20) | 0.1432 | 0.8075 | 0.1661 | 0.6579 | 0.5901 | 0.6136 | 0.4203 | 0.5225 |
| **DP-MERF** (AISTATS'21) | 0.6367 | 0.6738 | 0.5862 | 0.6162 | 0.5936 | 0.6082 | 0.4413 | 0.4489 |
| **P3GM** (ICDE'21) | 0.7369 | 0.7981 | 0.7223 | 0.7480 | 0.5673 | 0.5884 | 0.4532 | 0.4858 |
| **G-PATE** (NeurIPS'21) | 0.5810 | 0.8092 | 0.5567 | 0.6934 | 0.6702 | 0.6897 | 0.4985 | 0.6217 |
| **DataLens** (CCS'21) | 0.7123 | 0.8066 | 0.6478 | 0.7061 | 0.7058 | 0.7287 | 0.6061 | 0.6224 |
| **DPGEN** (CVPR'22) | 0.9046 | 0.9357 | 0.8283 | 0.8784 | 0.6999 | 0.8835 | 0.6614 | 0.8147 |
| **DPSH** (NeurIPS'21) | N/A | 0.8320 | N/A | 0.7110 | N/A | N/A | N/A | N/A |
| **PSG** (NeurIPS'22) | 0.8090 | 0.9560 | 0.7020 | 0.7770 | N/A | N/A | N/A | N/A |
| **DPAC** (CVPR'23) | N/A | 0.8800 | N/A | 0.7300 | N/A | N/A | N/A | N/A |
| *With pre-training* | | | | | | | | |
| **DP-DM** (TMLR'23) | 0.9520 | 0.9810 | 0.7940 | 0.8620 | 0.7108 | 0.8586 | 0.7513 | 0.8018 |
| **DPGU** (arXiv'23) | N/A | **0.9860** | N/A | N/A | N/A | N/A | N/A | N/A |
| **DP-LDM** (arXiv'23) | 0.9590 | 0.9740 | 0.7890 | 0.8514 | 0.6572 | 0.8417 | 0.6851 | 0.7846 |
| **PGE** (Ours) | **0.9612** | 0.9751 | **0.8359** | **0.8934** | **0.7321** | **0.8983** | **0.7153** | **0.8401** |

**Baselines.** We compare our PGE with 14 state-of-the-art approaches, including 7 DPSGD-based approaches (DP-GAN [56], DP-MERF [25], GS-WGAN [9], P3GM [48], PSG [8], DPSH [6], DP-DM [12], DPGU [22], DPAC [7] and DP-LDM [39]), 3 PATE-based approaches (PATE-GAN [31], G-PATE [38] and DataLens [49]) and DPGEN [10] based randomized response. We get the experimental results from their original papers or run their official codes.

**Implementations.** We choose the same UNet as DP-DM to fit the potential energy of the system. When comparing classifier performance, we choose the same architecture as the other baselines. We initialize the network and the classifier using Kaiming initialization. If not emphasized, we set $k$ to 10 by default and $c$ to sample from a Gaussian distribution by default. The training epoch of network is 10,000 for MNIST, FashionMNIST and 50,000 for CelebA, LSUN. For each dataset, we generate 10,000 samples for classifier learning. The initial value of step size $\lambda$ is $10^{-5}$ and the sampling epoch is 1,000. We perform Metropolis Guidelines to decide whether to accept every 100 epochs of sampling.

**Metrics.** We evaluate our PGE as well as baselines in terms of classification accuracy and perceptual scores under the same different privacy budget constraints. In particular, the classification accuracy is evaluated by training a classifier with the generated data and testing it on real test datasets. Perceptual scores are evaluated by Inception Score (IS) and Frechet Inception Distance (FID), which are standard metrics for the visual quality of images.

### 5.2 Experimental Results

**Classification accuracy comparisons.** In order to demonstrate the effectiveness of our approach, we compare it with 11 state-of-the-art baselines under two privacy budget setting $\varepsilon = 1$ and $\varepsilon = 10$ on MNIST, FMNIST, CelebA-H and CelebA-G. Both our PGE and DPGEN satisfy pure DP, while the other baselines have a failure probability $\delta = 10^{-5}$. We evaluate the classification accuracy of the classifiers trained on the generated data, and the results are summarized in Tab. 1. It is important to note that our approach does not require pre-training with any dataset. Compared to approaches

**Table 2: Perceptual scores comparisons with 9 state-of-the-art baselines on CelebA at $64 \times 64$ resolution under different privacy budget $\varepsilon$.**

| Approach | $\varepsilon$ | IS↑ | FID↓ |
|---|---|---|---|
| *Without pre-training* | | | |
| **DP-GAN** (arXiv'18) | $10^4$ | 1.00 | 403.94 |
| **PATE-GAN** (ICLR'19) | $10^4$ | 1.00 | 397.62 |
| **GS-WGAN** (NeurIPS'20) | $10^4$ | 1.00 | 384.78 |
| **DP-MERF** (AISTATS'21) | $10^4$ | 1.36 | 327.24 |
| **P3GM** (ICDE'21) | $10^4$ | 1.37 | 435.60 |
| **G-PATE** (NeurIPS'21) | 10 | 1.37 | 305.92 |
| **DataLens** (CCS'21) | 10 | 1.42 | 320.84 |
| **DPGEN** (CVPR'22) | 10 | 1.48 | 55.910 |
| *With pre-training* | | | |
| **DP-LDM** (arXiv'23) | 10 | N/A | 14.300 |
| **PGE** (Ours) | 10 | **2.14** | **12.583** |

without pre-training, we observe consistent and significant improvements of around 4-6% across different configurations. This improvement is attributed to the fact that most approaches without pre-training rely on GANs and Gaussian mechanism for differentially private generative modeling. In contrast, our approach combines a more stable EBM with RR to achieve a better balance between privacy and data utility. Furthermore, when compared to two DDPM-based approaches with pre-training, our approach consistently achieves optimal results in most settings. This can be attributed to the fact that EBMs converge with fewer queries compared to DDPM, resulting in better performance. These results suggest that our PGE can effectively generate high-resolution images with practical applications.

**Perceptual scores comparisons.** To further demonstrate the effectiveness of our approach, we evaluate it using two metrics: IS and FID, as mentioned earlier. Since there are no corresponding experimental results for PSG and DP-DM, we compared our approach with the remaining 9 approaches. The results are shown in Tab. 2. Here, a superior IS value signifies enhanced quality of the generated samples, whereas a lower FID score indicates a closer

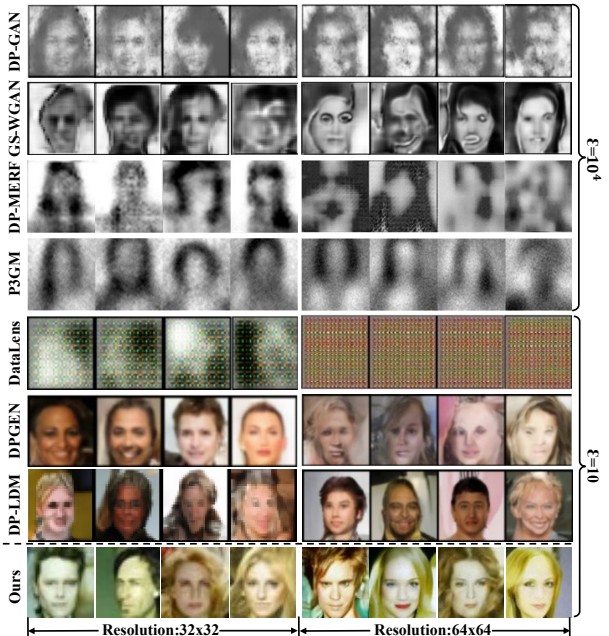

Figure 4: Visualization results of DP-GAN, GS-WGAN, DP-MERF, P3GM, DataLens, DPGEN, DP-LDM and our PGE on CelebA at 32×32 and 64×64 resolutions.

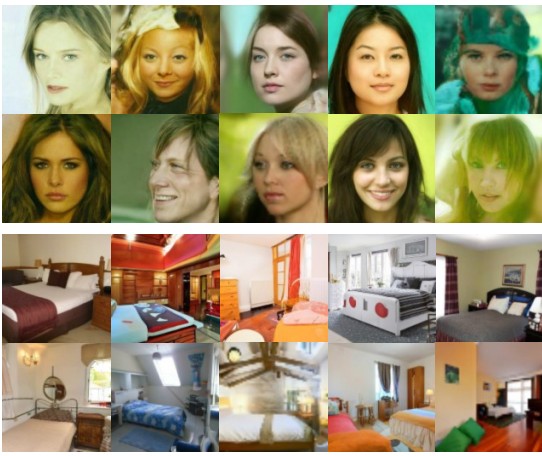

Figure 5: Visualization results of CelebA and LSUN at 256×256 resolution under $\varepsilon = 20$.

resemblance to authentic images. Our approach outperforms the other baseline approaches by achieving the highest IS of 2.14 and the lowest FID of 12.583, particularly notable under the most stringent privacy budget of 10. This can be attributed to two main factors. Firstly, we achieve differential privacy by employing RR instead of the Gaussian mechanism, which helps to avoid direct damage to the gradients. Secondly, EBMs exhibit better stability during the training process compared to GANs. These results emphasize the robustness of our approach. Despite the randomized response perturbation, the trained network can still accurately predict the positions of the realistic images.

**Visual comparisons of generated data.** Furthermore, we provide visual evidence to demonstrate the excellent quality of the generated data by our approach. We compare the visualization results with other baselines in Fig. 4. Even under a high privacy budget condition of $\varepsilon = 10^4$, the grayscale images generated by DP-GAN, GS-WGAN, DP-MERF, and P3GM are still blurry in comparison. Grayscale images have lower dimensionality compared to color images, which makes it relatively easier to balance data quality with privacy protection. The color images generated by DPGEN and DP-LDM exhibit better visual quality than DataLens but lack fleshed-out facial details. In comparison, the images generated by our PGE appear more realistic and have more complete facial details, further confirming the effectiveness of our approach.

**Visualization for images with 256 resolution.** To further elucidate the capacity of our PGE to generate high-resolution images, we conduct visualizations of images at a resolution of 256, under a setting of $\varepsilon = 20$. The results are presented in Fig. 5, where the

generated images serve as empirical evidence to the efficacy of our approach. Notably, despite the augmented complexity inherent to processing images of a 256×256 resolution, our PGE exhibits a robust ability to accurately model the underlying data distribution. This is a critical observation, as it indicates the preservation of essential visual features even at higher resolutions. Moreover, a detailed comparison with images of lower resolutions (those less than 256) reveals a significant enhancement in the richness of detail within these higher-resolution images. Such improvement is not merely cosmetic but has substantial implications for the utility of the generated data. Specifically, the enhanced detail facilitates the deployment of these images in more demanding downstream tasks, where fine-grained visual information is pivotal. Thus, the results highlight the effectiveness of our approach with respect to the need for high-resolution image generation.

### 5.3 Ablation Studies

After the promising performance is achieved, we further analyze the impact of each component of our approach, including the hyperparameter $k$, the step size $\lambda$ and the initial distribution of kinetic energy $K(c)$.

**Impact of $k$.** To investigate the impact of the hyperparameter $k$ on the trade-off between privacy and data utility, we compare the perceptual scores obtained when $k$ takes different values under the same privacy budget $\varepsilon = 10$. To make the experimental results more representative, we chose two datasets, CelebA and LSUN, with a resolution of 256×256. The results are presented in Fig. 6. As $k$ increases, the FID score increases, indicating a decrease in the quality of the generated image. This aligns with our expectations. According to Eq. (6), a smaller $k$ value increases the likelihood of $\mathcal{R}(v_i) = v_i$, which in turn enhances the probability that the model accurately captures the underlying data manifold. This implies that the generated data will resemble the distribution of private data more closely. Therefore, as $k$ increases, there is a slight decline in the quality of the generated images.

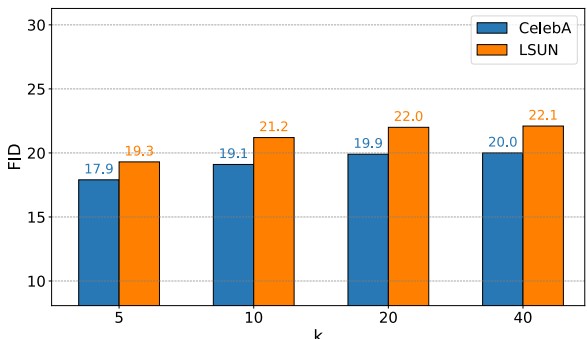

**Figure 6: FID score on LSUN and CelebA at 256×256 resolution under different $k$.**

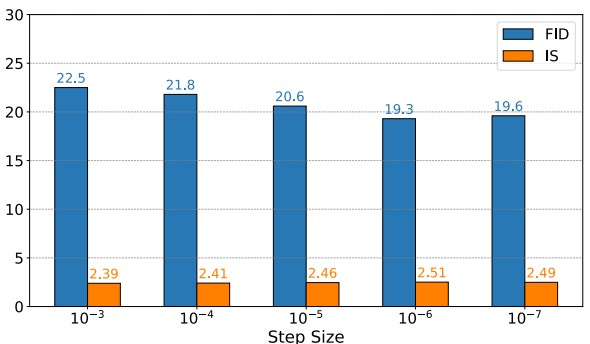

**Figure 7: Perceptual scores on LSUN at 256×256 resolution under different step size $\lambda$.**

**Impact of step size.** To study the effect of the step size $\lambda$ on the quality of the generated images, we generate images with different step size $\lambda$ and compare their perceptual scores. The results are shown in Fig. 7. Our findings suggest that decreasing the step size $\lambda$ weakly improves the quality of the generated images when $\lambda$ is below $1e − 6$. However, at $\lambda = 1e − 7$, we observe a slight decrease in the quality of the generated images compared to $\lambda = 1e − 6$. Similar to the learning rate in machine learning, a smaller $\lambda$ leads to finer adjustments per sample, potentially enriching image detail. Nevertheless, excessively small $\lambda$ values may cause the generation process to converge to local optima, compromising the quality of the resulting images. Therefore, achieving the best balance in choosing $\lambda$ is critical to enhancing image detail while ensuring that it doesn't get in the way of other optimization workflows.

**Impact of kinetic energy distribution.** To explore the effect of the initial distribution of kinetic energy $K(c)$ (or $c$) on the data quality. Images are sampled with different initial distribution (Gaussian, Rayleigh, and Uniform) of $c$ and show the results in Fig. 8. It is observed that the images generated with the initial value of $c$ sampled from the Gaussian distribution exhibit the highest quality. The Rayleigh distribution, which is the joint distribution of two independent Gaussian distributions, yielded slightly lower-quality images than the Gaussian distribution. The lowest image quality is

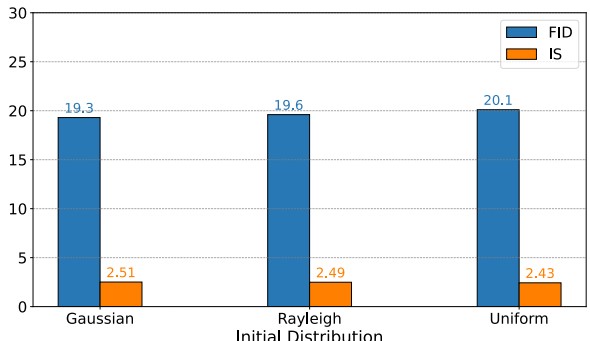

**Figure 8: Perceptual scores on LSUN at 256×256 resolution under different kinetic energy distribution $K(c)$.**

observed when the initial values of $c$ are sampled from Uniform distribution, but the difference in quality between the images obtained when the initial value of $c$ was sampled from the three distributions is not very large. The conservation of both kinetic and potential energy in a system not affected by external forces is the main reason for this. Additionally, Eq. (11) reveals that there is an interaction between the two energies, and that the varying distributions only impact the initial energy magnitude of the system.

## 6 Limitations

There are still some limitations to PGE. We summarize them here. (1) Our use of Hamiltonian dynamics for sampling, unlike GANs that generate images all at once, requires iterative querying, making it significantly slower, especially for high-dimensional images—up to a hundred times slower than GANs. (2) The generated images often lack detailed backgrounds; for instance, CelebA images typically have a solid color background. (3) Our approach struggles with class information extraction compared to classifier-guided diffusion models, as embedding labels directly into images lacks theoretical support. Developing a class-guided sampling approach is a future goal. (4) The model may inadvertently learn dataset biases, which we aim to address through data preprocessing in future work.

## 7 Conclusion

Releasing private data or trained networks can lead to privacy leakage. To ensure secure deployment, we propose a PGE approach that generates privacy-preserving images for various tasks. By integrating the RR mechanism into the training of EBMs, our approach balances privacy and utility more effectively than other state-of-the-art approaches. Additionally, our MCMC sampling algorithm based on Hamiltonian dynamics enhances realistic data generation. Moreover, we conducted detailed privacy and convergence analyses for our PGE. Notably, PGE satisfies pure DP, eliminating the failure probability present in most other DP generative approachs. Extensive experiments and privacy and convergence analysis are conducted to show the effectiveness and rationality of our approach.

**Acknowledgements.** This work was supported by grants from the Pioneer R&D Program of Zhejiang Province (2024C01024).

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
