# OpenReview forum: "Private Gradient Estimation is Useful for Generative Modeling"
_acmmm.org/ACMMM/2024/Conference — MM2024 Oral_

### Official Review · Reviewer_Lr9Z · 2024-05-24

**Rating:** 5
**Confidence:** 2

**Summary:**

The paper introduces an approach to generate high-quality, high-resolution synthetic data while ensuring privacy guarantees. It builds generative models by incorporating differential privacy into the training process of EBMs. This method uses Hamiltonian dynamics and differentially private gradient estimation to generate high-resolution data. The proposed approach achieves differential privacy through the perturbation of projection vectors during gradient estimation with sliced score matching and incorporates a residual enhancement module for improved reconstruction ability. Experiments show the proposed method achieves SOTA performance.

**Strengths:**

The introduction of PGE combines Hamiltonian dynamics with differential privacy, offering a novel method for generating high-resolution, privacy-preserving images. The extensive experiments show that the approach achieves improvements in visual quality and data utility over state-of-the-art methods. The paper provides comprehensive privacy and convergence analyses, validating the theoretical foundations and practical effectiveness of the proposed approach. The paper is well-written.

**Limitations:**

(1)The combination of Hamiltonian dynamics, randomized response, and sliced score matching increases the complexity of implementation, which might be challenging for practitioners to reproduce. The iterative nature of MCMC sampling with Hamiltonian dynamics can be slow, particularly for generating large batches of high-resolution images, impacting the scalability of the approach.
(2) In experiments, the authors use large privacy budget epsilon, i.e., 𝜀= 1 and 𝜀 = 10. I would like to see results under a smaller privacy budget (𝜀< 1).
(3) For experiment comparison, the experiments of DP-DM and DP-LDM are conducted on MNIST and FMNIST datasets. It would be convincing to report experiments from DP-DM and DP-LDM on CelebA-H and CelebA-G datasets.
(3) There are redundant and repetitive descriptions in the introduction.
(4) In related work, in the Differentially Private Learning section, the sentence “This is mainly because existing works apply training methods for discriminative tasks directly to generative tasks” should be supported. Please discuss some reference works that applied techniques from DP discriminative tasks to generative tasks.
(5) In preliminaries in Energy-based models (EBMs), please add references/citations to the sentence “Notably, it is easy to extend to the multi-dimensional case as long as multiple variables are distributed independently of each other.”
(6) What does a releasable network mean? Is it the network that can generate data without any privacy leakage?
(7) In Table 1, on MNIST at the privacy budget of 10, your method is inferior to the DP-DM. Please provide the reason for it.
(8) In Table 2, all the algorithms should have the same privacy budget.
(9) does your approach show any disparity in under-represented data groups if the target data is skewed? If yes, what could be the possible measures to avoid it?

**Suitability:**

3

---

### Official Review · Reviewer_9zUG · 2024-05-25

**Rating:** 4
**Confidence:** 3

**Summary:**

The paper addresses privacy concerns in generative models by introducing a differentially private approach suitable for high-dimensional data generation. The method leverages Hamiltonian dynamics and gradient estimation via a well-trained network to generate samples. Privacy is ensured by perturbing projection vectors in the gradient estimation using sliced score matching. Additionally, a residual enhancement module improves the model's reconstruction ability. This approach enables the generation of 256×256 resolution images. Extensive experiments demonstrate the method's effectiveness and rationality compared to existing techniques.

**Strengths:**

+ This work offers a new private generative modeling approach where samples are generated via Hamiltonian dynamics with gradients of the private dataset estimated by a well-trained network.
+ Extensive case studies.
+ Detailed privacy analysis.

**Limitations:**

- The method performance needs to be further verified under small privacy budget.
- Convergence-related experiments need to be supplemented theoretically or empirically.
- The writing quality of this paper needs another round of polishing.


This paper is well organized and logically clear, especially designing a private gradient estimation method for generative models that achieves a good privacy-utility trade-off. Furthermore, the authors comprehensively validate the proposed method theoretically and empirically. However, I still have the following minor issues that need to be addressed by the authors:

1.	The method performance needs to be further verified under small privacy budget. I want to know how the proposed method performs with a small privacy budget. For example, in DataLens, the authors conducted a similar exploration, which is beneficial for understanding the performance bounds of the proposed method.

2.	Convergence-related experiments need to be supplemented theoretically or empirically. The authors mentioned in the limitations that the time overhead of the proposed method is expensive, then I would like to know how effective the convergence of the proposed method is. Therefore, it would be better if the authors could analyze the convergence of the proposed method empirically or theoretically.

3.	Furthermore, the authors claim that the background of the generated CelebA dataset lacks details, but in data quality assessment experiments, the quality of the data generated by the proposed method is very high, whether this is a contradictory statement.

4.	In table 2, I would like to know what is the quality of real data. In DataLens, the authors report the quality of real data, which is useful for comparing the performance of different methods in terms of generation quality.

Minor issues:

-	There are many typos and grammatical errors in the context of this article, for example, the text on line 101 exceeds the text box.

**Suitability:**

2

---

### Official Review · Reviewer_W6Bq · 2024-05-25

**Rating:** 4
**Confidence:** 2

**Summary:**

The paper presents an approach to differentially private generative modeling by addressing the challenge of generating high-dimensional data while preserving privacy, which existing methods struggle with due to the complexity and instability of current models like GANs and diffusion models.

**Strengths:**

- The paper provides detailed mathematical formulations and justifications for using Fisher divergence to train EBMs
- The PGE approach is theoretically sound in its application of the RR mechanism to achieve (ε, 0)-differential privacy
- The method is thoroughly evaluated on four standard datasets with varying resolutions, with comprehensive ablation study

**Limitations:**

- Hamiltonian dynamics-based sampling is computationally intensive, especially for high-dimensional data, potentially could be a major drawback for practical applications

**Suitability:**

3

---

### Official Review · Reviewer_N8jx · 2024-05-29

**Rating:** 4
**Confidence:** 3

**Summary:**

This paper introduces a new privacy-preserving generative model that incorporates differential privacy into the training of energy-based models. It also integrates a residual enhancement module inspired by masked autoencoders and employs a sampling method based on Hamiltonian dynamics. As a result, the method outperforms existing GAN-based and diffusion-based private generative models. Experimental results are presented in terms of downstream classification accuracy, IS, and FID.

**Strengths:**

1. The paper is well-structured and easy to follow.
2. The paper addresses a well-motivated and widely-investigated task: training private generative models, which is significant for protecting training data's privacy.
3. Previous methods in this field were mainly focusing on incorporating differential privacy with GANs and Diffusion Models, while this paper addresses the task using energy-based models, which increases the diversity of the field.
4. The results look promising, as it surpasses all baseline methods on utility and privacy criteria.

**Limitations:**

1. As the residual enhancement module is a key component of the method, it is recommended to also conduct its ablation study to verify its contribution, and also to see how the method performs without this module.
2. There are numerous existing methods in this field. Although the proposed method has been compared with a wide range of baselines, for a more comprehensive evaluation that convincingly demonstrates its superior performance, the authors should consider including the current best-performing diffusion-based method [1] and GAN-based method [2], which respectively improve upon the DPDM [3] and GS-WGAN [4] currently used for comparison. Additionally, to enhance the diversity of the comparison, another work [5] that is not based on GANs or diffusion models could also be included.

[1] Differentially Private Diffusion Models Generate Useful Synthetic Images. arXiv, 2023.

[2] Private Image Generation with Dual-Purpose Auxiliary Classifier. CVPR, 2023.

[3] Differentially Private Diffusion Models. TMLR, 2023.

[4] GS-WGAN: A Gradient-Sanitized Approach for Learning Differentially Private Generators. NeurIPS, 2020.

[5] Don't Generate Me: Training Differentially Private Generative Models with Sinkhorn Divergence. NeurIPS, 2021.

**Suitability:**

3

---

### Meta-Review · Area_Chair_kVj1 · 2024-07-02

**Recommendation:** Accept (Oral)
**Confidence:** 5

**Metareview:**

The reviewers agreed that this work was well-written, included extensive experiments/case-studies, had a detailed privacy analysis, was well justified from a mathematical perspective. N8jx notes that while it “addresses a well-motivated and widely-investigated task”, it takes a different approach that “increases the diversity of the field” supporting that this is a novel contribution.

In terms of limitations, N8jx notes an extra ablation study that could be included and comparison with some extra methods. These issues were addressed in the rebuttal. W6Bq notes that this method may be computationally intensive, a potential drawback. 9zUG notes some “minor issues” related to verifying performance with a small privacy budget, convergence experiments, and related to data quality. Lr9Z echoes some of these issues related to the privacy budget, and highlights potential issues with reproduction.

The overall scores are positive with all reviewers recommending acceptance. Given that 3/4 reviewers also suggest that this work is “definitely suitable” for the MM community, I will add my vote for acceptance as well provided that the updates in the rebuttal are incorporated into the manuscript.